# The Prognostic Value of the Systemic Immune-Inflammation Index (SII) and Red Cell Distribution Width (RDW) in Patients with Cervical Cancer Treated Using Radiotherapy

**DOI:** 10.3390/cancers16081542

**Published:** 2024-04-18

**Authors:** Emilia Staniewska, Karolina Grudzien, Magdalena Stankiewicz, Katarzyna Raczek-Zwierzycka, Justyna Rembak-Szynkiewicz, Zuzanna Nowicka, Rafal Tarnawski, Marcin Miszczyk

**Affiliations:** 1IIIrd Department of Radiotherapy and Chemotherapy, Maria Sklodowska-Curie National Research Institute of Oncology, 44-102 Gliwice, Poland; 2Radiotherapy Department, Maria Sklodowska-Curie National Research Institute of Oncology, 44-102 Gliwice, Poland; 3Brachytherapy Department, Maria Sklodowska-Curie National Research Institute of Oncology, 44-102 Gliwice, Poland; 4Radiology and Diagnostic Imaging Department, Maria Sklodowska-Curie National Research Institute of Oncology, 44-102 Gliwice, Poland; 5Department of Biostatistics and Translational Medicine, Medical University of Lodz, 90-419 Lodz, Poland; 6Department of Urology, Comprehensive Cancer Center, Medical University of Vienna, 1090 Vienna, Austria; 7Collegium Medicum—Faculty of Medicine, WSB University, 41-300 Dąbrowa Górnicza, Poland

**Keywords:** cervical cancer, radiotherapy, chemoradiation therapy, systemic immune-inflammation index, red cell distribution width, overall survival

## Abstract

**Simple Summary:**

Cervical cancer remains the fourth leading cause of cancer-related deaths among women worldwide. The red cell distribution width (RDW) and systemic immune-inflammation index (SII) are common, well-known haematological indices. The aim of this retrospective study was to evaluate the association between pre-treatment RDW and SII, and overall survival (OS) in 249 patients treated with definitive chemoradiation therapy (CRT) for histopathologically confirmed, primary localised cervical cancer. Statistical analysis was performed using the Kaplan–Meier method, two-sided log-rank tests, and Cox proportional hazards models, with the Akaike Information Criterion (AIC) serving as a prediction error estimator. The vast majority of patients (95.2%) were diagnosed with squamous cell carcinoma (SCC) in FIGO stage III (84.7%). Patients with a low RDW (≤13.4%) and low SII (≤986.01) had a significantly longer OS (*p* < 0.01). The RDW remained as an independent prognostic factor in the multivariable model (*p* < 0.01). The RDW is a cheap and easily accessible index that could be used to improve pre-treatment prognosis assessments in patients with cervical cancer undergoing CRT.

**Abstract:**

Introduction: There is growing interest in the prognostic value of routinely performed pre-treatment blood test indices, such as the RDW or SII, with the latter combining the neutrophil-to-lymphocyte ratio (NLR) and platelet-to-lymphocyte ratio (PLR). These indices were shown to be prognostic for survival in some malignancies. The purpose of this study was to evaluate the association between pre-treatment RDW and SII, and OS in patients treated with radiotherapy for primary localised cervical cancer. Material and Methods: This retrospective analysis included patients treated with definitive CRT between 2011 and 2017 for histopathologically confirmed FIGO 2018 stage IB2-IVA cervical cancer. Statistical analysis was performed using the Kaplan–Meier method, two-sided log-rank tests, and Cox proportional hazards models, with the AIC serving as a prediction error estimator. Results: The study group included 249 patients with a median age of 57.2 years and a median follow-up of 75.8 months. The majority were diagnosed with squamous cell carcinoma (237; 95.2%) and had FIGO stage III (211; 84.7%). Approximately half of the patients (116; 46.4%) had regional lymph node metastases. Patients with a low RDW (≤13.4%) and low SII (≤986.01) had a significantly longer OS (*p* = 0.001 and *p* = 0.002). The RDW remained as an independent prognostic factor in the multivariable model (high vs. low; HR = 2.04; 95% CI: 1.32–3.16; *p* = 0.001). Including RDW in the model decreased the Akaike Information Criterion from 1028.25 to 1018.15. Conclusions: The RDW is a cheap and widely available index that is simultaneously an independent prognostic factor for survival and could be used to improve pre-treatment prognosis assessments in patients with cervical cancer undergoing CRT. Available data encourage assessing the RDW as a prognostic factor in prospective trials to aid the identification of candidates for treatment escalation.

## 1. Introduction

Each year, approximately 2360 new cases of cervical cancer are diagnosed in Poland, and 570,000 are diagnosed globally [1,2]. Despite the introduction of screening programmes that reduce mortality by at least 80% [1], cervical cancer remains the eighth primary cause of cancer-related death among women in Poland [2] and the fourth worldwide [3]. According to the current guidelines, patients with the International Federation of Gynaecology and Obstetrics (FIGO) [4] stage IB2–IVA disease are most often offered concurrent chemoradiation therapy (CRT) combined with a brachytherapy (BT) boost [5]. The 5-year overall survival (OS) varies from 15 to 95%, depending on the initial FIGO stage (80–95% for IB2/IIA, 70–85% for IIB, 40–65% for III, and 15–25% for stage IVA) [5].

In addition to the FIGO stage, specific blood tests, such as elevated serum squamous cell carcinoma antigen (SCCA), have demonstrated prognostic value for patients with locally advanced cervical carcinoma undergoing definitive CRT [5]. The red cell distribution width (RDW), neutrophil-to-lymphocyte ratio (NLR), and platelet-to-lymphocyte ratio (PLR) are pre-treatment blood test indices that may be associated with clinical outcomes. The RDW represents the red blood cell (RBC) volume variation. Elevated values are associated with bone marrow dysfunction [6] and inflammation related to cancer [7] or non-oncological diseases [8,9,10].

Neutrophil-to-lymphocyte ratio (NLR) and platelet-to-lymphocyte ratio (PLR) can be combined to generate the so-called systemic immune-inflammation index (SII), which is defined as follows: $SII=\frac{\left[neutrophils\;\times \;platelets\right]}{\left[lymphocytes\right]}$.

The RDW and SII have demonstrated a substantial prognostic negative value in several malignancies, including colorectal cancer [11], prostate cancer [12], gynaecological tumours [13,14,15], head and neck cancers [16,17,18,19], Hodgkin lymphoma [20], and gliomas [21]. Both indices reflect the inflammatory response related to the accelerated cancerogenesis and neo-angiogenesis induced by the cytokines produced by cancer cells [22]. Importantly, due to their availability via routine blood tests, both the RDW and SII could potentially be incorporated into risk stratification without incurring additional costs.

We sought to ascertain whether the RDW and SII could be used as independent prognostic factors for survival in patients with cervical cancer treated with definitive CRT.

## 2. Material and Methods

### 2.1. Study Group

This retrospective study included patients treated for histopathologically confirmed cervical cancer with definitive CRT between 2011 and 2017 at a tertiary institution. The inclusion criteria were as follows: FIGO 2018 stage IB2-IVA, pre-treatment diagnostic imaging including at least ^18^F-fluorodeoxyglucose positron emission tomography-computed tomography (PET-CT), and availability of pre-treatment blood tests. The exclusion criteria were palliative intent of treatment or prior surgical treatment for cervical carcinoma.

### 2.2. Methodology

The results of blood test performed before CRT were extracted from institutional medical records. All blood tests were conducted in the same laboratory. SII was determined by applying the formula mentioned previously.

TNM and FIGO staging was re-evaluated and updated using the American Joint Committee on Cancer (AJCC) 8th Edition (2017) [23] and 2018 FIGO system [4], including a retrospective re-assessment of diagnostic imaging performed prior to the initiation of treatment: magnetic resonance imaging (MRI), computed tomography (CT), and PET-CT.

Survival data were obtained from the Polish National Cancer Registry. As the primary endpoint, overall survival was defined as the time between the first fraction of external beam radiotherapy (EBRT) or the first dose of concurrent chemotherapy (CTx) and the date of death. In all applicable cases, dates of patients’ deaths were available, and the remaining cases were censored using the last known date on which the patient was alive.

The follow-up (FU) was generally conducted in accordance with institutional guidelines. FU visits were scheduled every two to three months in the first year, every three to four months in the second year, biannually throughout the third and fourth years, and annually thereafter. FU visits included a gynaecological examination and laboratory testing, such as baseline blood tests and tumour markers: SCCA, carcinoma antigen 125 (CA-125), and carcinoembryonic antigen (CEA). The first MRI was typically performed six months after CRT, unless clinically indicated earlier, and CT was performed annually.

The institutional Bioethical Committee (Maria Sklodowska-Curie National Research Institute of Oncology, Gliwice, Poland, KB/430-81/21 on 29 June 2021) approved the study protocol.

### 2.3. Statistical Analysis

Statistical analysis was conducted using the Kaplan–Meier method, two-sided log-rank tests, and Cox proportional hazards models. The optimal cut-off values for continuous predictors were identified as those corresponding to the most significant relationships with the outcome (survival). In each case, where applicable, the relationship between continuous and dichotomized (high vs. low) predictors and survival was reported.

Survival analysis included known clinical factors—such as age, histopathology, TNM and FIGO stage groups, and ECOG (Eastern Oncology Group) performance status—and investigated parameters—including the RDW, NLR, PLR, and SII. Hazard ratios with 95% confidence intervals (CIs) were reported for all variables. If multiple interdependent variables were statistically significant (i.e., NLR, PLR, and SII), only the most statistically significant variables were included in the multivariable analysis. The Akaike Information Criterion (AIC) was used as a prediction error estimator.

The correlations between the RDW and SII, and the FIGO stage and ECOG performance status, were analysed using Spearman’s R coefficient and non-parametric Kruskal–Wallis ANOVA, respectively.

The statistical analysis was conducted using TIBCO Software Inc.’s (Palo Alto, CA, USA) STATISTICA 13.3, the Survminer R package (version 0.4.9), and the survival package (version 3.3-1). Plots were generated using ggplot (version 2_3.3.5). *p* values of less than 0.05 were considered statistically significant.

## 3. Results

### 3.1. Study Group Description

The final analysis included 249 patients. Thirty-eight cases were excluded from the initial database due to prior surgical treatment, palliative intent of the treatment, or FIGO stage IVB. The median age at diagnosis was 57.2 (interquartile range, IQR: 49.2–64.5). The majority of patients were diagnosed with squamous cell carcinoma (237, 95.2%) and FIGO stage III (211, 84.7%). In 116 (46.4%) cases, metastatic pelvic and/or para-aortic lymph nodes were positive. The clinical characteristics of the study group are presented in Table 1.

Treatment included EBRT (249, 100%), subsequent BT in 246 patients (98.8%), and concurrent CTx with Cisplatin (40 mg/m^2^ weekly) as the solitary agent in 225 patients (90.0%). The median EQD2 (equivalent dose), including EBRT and BT, in the group was 84.35 Gy (IQR 80.81–106.91 Gy). Appendix A provide a comprehensive description of radiotherapy schedules. Neither lumboaortic irradiation, nor bone marrow sparing was performed. All patients received elective pelvic lymph node irradiation either up to the level of L4/L5 vertebral junction or up to the aortic bifurcation, depending on the attending physician’s decision. In two cases, BT was omitted due to technical considerations, and, in one case, the patient withdrew consent. Twenty-four patients (9.6%) were excluded from CTx due to concomitant disease burden (10, 4%), blood tests (3, 1.2%), a history of Cisplatin-related hypersensitivity (1, 0.4%), advanced age and low performance status (7, 2.8%), or unknown causes (3, 1.2%). CTx was discontinued in five patients (2%) after the first cycle due to adverse effects (kidney failure, bradycardia, and leukopenia).

All patients had complete census-based survival data, with a median observation time of 75.8 months (IQR 67.83–76.06). At the time of analysis, 60.2% (150) of the patients were still alive. The median OS for the entire study cohort has not yet been reached.

### 3.2. Prognostic Value of RDW and SII

Figure 1A,B depict the cut-off values, which were determined to be 13.4% for the RDW and 986.01 for the SII.

Patients with a low RDW (≤13.4% vs. >13.4%; *p* = 0.001) and low SII (≤986.01 vs. >986.01; *p* = 0.002) had a significantly higher OS. Figure 2A depicts that the median OS for patients with RDW > 13.4% being 65.8 months, while it was not reached for those with RDW ≤ 13.4%. Similarly, the median OS for patients with an SII > 986.01 was 65.8 months, whereas the median OS for patients with an SII ≤ 986.81 was not reached (Figure 2B).

Table 2 displays the results of the univariate and multivariable Cox regression analyses. If multiple interdependent variables were statistically significant (i.e., NLR, PLR, and SII), only the most statistically significant variables were included in the multivariable analysis (SII instead of NLR and PLR, and FIGO stage instead of TNM stage). In the multivariable model, RDW > 13.4 (HR = 2.04, 95% CI 1.32–3.16, *p* = 0.001) and ECOG status = 2 (HR = 4.55, 95% CI 1.61–12.85, *p* = 0.023) remained as independent prognostic factors for survival.

The Akaike Information Criterion decreased from 1028.25 to 1018.15 when blood test indices were incorporated into the clinical model, indicating a better fit of the model.

Potential clinically significant correlations were performed between the RDW and SII, and the FIGO stage and ECOG performance status. The SII was significantly correlated with the FIGO stage (Spearman’s R = 0.32, *p* < 0.001) in contrast to the RDW (*p* = 0.094). No correlations were found between the SII (*p* = 0.078) nor the RDW (*p* = 0.966) or ECOG performance status.

## 4. Discussion

Despite a growing body of evidence, pre-treatment blood parameters are rarely incorporated into the assessment of a patient’s prognosis. Several indices, including the eosinophil/lymphocyte ratio (ELR) [24], NLR, PLR, thrombocyte-to-lymphocyte ratio (TLR), C-reactive protein/albumin ratio (CAR) [25], or SII [13,24,26], have been proposed in the literature. Given that the expected survival rate for patients who are CRT-treated ranges between 15% and 95% [5], enhancements to the pre-treatment evaluation are essential. Adding blood test indices to the clinical model enhances the prognostic value, and the RDW could be used to improve the accuracy prognosis assessment of patients with cervical cancer treated using CRT.

Although this is a retrospective study, the number of patients included is relatively high, with a long duration of follow-up; moreover, the majority of control visits were conducted in accordance with institutional protocols, thereby reducing bias. In addition to histopathological diagnoses, diagnostic imaging and treatment options were also available. Uniquely, the stage of disease was re-assessed in accordance with FIGO 2018 [4] and AJCC 8th Edition guidelines [23]. Importantly, we were able to obtain comprehensive survival data, and each patient’s FU exceeded 5 years. Our study group was also relatively non-heterogeneous as each patient was evaluated for definitive CRT, whereas numerous treatment modalities were used in the other study groups [24,27,28].

Holub et al. [24] conducted a study with 151 (FIGO stage IA–IVB) patients, of whom only 85 (60.7%) underwent CRT with subsequent BT. SII ≥ 1000 was only associated with poorer survival in the univariate Cox regression analysis that also accounted for NLR, PLR, neutrophils, FIGO stage III–IV, bulky tumour, ELR, eosinophils, and age. Only ELR remained as an independent survival prognostic factor following the multivariable analysis of blood test indices.

Huang et al. [13] analysed 458 patients with FIGO IA-IIA stage cervical cancer. Patients were divided into a high-SII group (SII > 475) and a low-SII group (SII ≤ 475). The high SII > 475 was associated with significantly worse survival in univariate (HR = 2.46, 95% CI 1.52–3.96, *p* < 0.001) and multivariable models (HR = 2.53, 95% CI 1.32–4.83, *p* = 0.005), and 5-year OS in the high-SII group was significantly shorter than in the low-SII group (*p* < 0.001).

Liu et al. [26] sought to determine SII’s capacity to predict patient responses to chemotherapy and long-term prognosis in 210 patients with cervical squamous cell carcinoma (CSCC) in FIGO stages IB2–IIB who underwent platinum-based neoadjuvant chemotherapy before surgery. The optimal SII cut-off values for a complete pathological response (pCR) and survival were calculated to be 568.7051 and 600.5683, respectively. A high SII was significantly associated with a decreased pCR, PFS (progression-free survival), and OS.

Ferioli et al. [29] presented the results of the ESTHER study. The aim of their research was to correlate the prognostic impact in LACCs of different pre-treatment nutritional and systemic inflammation indices, such as SII, on the following clinical endpoints: local control (LC), distant metastasis free survival (DMFS), DFS, and OS. Interestingly, the study group was quite similar to ours—173 cases were included and patients diagnosed with primary localised cervical cancer were treated with CRT, including EBRT and BT, with concurrent Cisplatin administered weekly. LACCs were retrospectively classified according to the 2018 FIGO staging system. According to the results, the SII was an independent prognostic factor in multivariable analysis for DFS (*p* < 0.01), but not for OS.

Guo et al. [30] conducted an analysis including 196 LACC patients treated with concurrent CRT (Paclitaxel and Cisplatin), and the authors took into account the prognostic value of body composition and systemic inflammatory markers, which is also an SII. Among the other factors, this marker was significantly associated with OS (*p* = 0.004) in a multivariable model, with a cut-off value of ≥1377.37.

In the majority of studies [27,28,31], there was no association between the RDW and survival in patients with cervical cancer. RDW > 14.66% was independently associated with a shorter PFS and OS only in one cohort study, which involved 440 patients with FIGO stage I–III cervical cancer treated with radical radiotherapy [32].

The numerous RDW (or SII) cut-off values reported by the authors do not facilitate the selection of a single value for clinical application. In addition to considering the comorbidities and ethnicity of patients, multicentre prospective external validation appears to be a viable solution.

There are several limitations to our study. The data were collected retrospectively, and the results are susceptible to associated biases. The majority of cases lacked comprehensive information regarding comorbidities, smoking behaviours, reproductive history, and a detailed histopathological report. In addition, it was impossible to account for the variety of EBRT and BT fractionation schemes used to treat the patients. Nevertheless, we believe that our findings contribute to the existing body of evidence.

## 5. Conclusions

A high RDW is associated with worse prognosis in patients with cervical cancer treated with chemoradiotherapy. It is a cheap and widely available index, which could be used to improve the accuracy of pre-treatment prognostic assessments to allow for treatment intensification in patients at high risk of failure. On the contrary, we did not find a significant association between the SII and survival. Prospective trials are necessary to verify the value of the RDW in improving decision making for patients with cervical cancer.

## Figures and Tables

**Figure 1 cancers-16-01542-f001:**
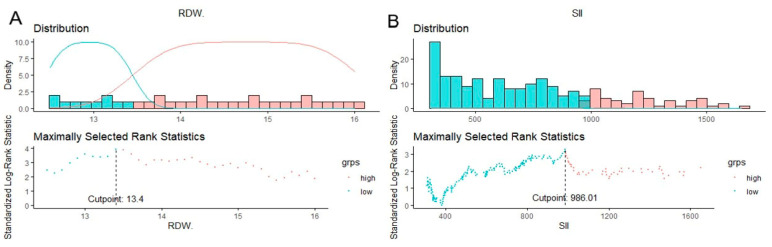
Cut-off point of RDW (**A**) and SII (**B**) selection using standardized log-rank statistics of 249 patients receiving definitive chemoradiation therapy for cervical cancer. Density means representation of distribution of RDW/SII; a mixture model of Gaussian distributions is fitted to the histogram of the biomarker.

**Figure 2 cancers-16-01542-f002:**
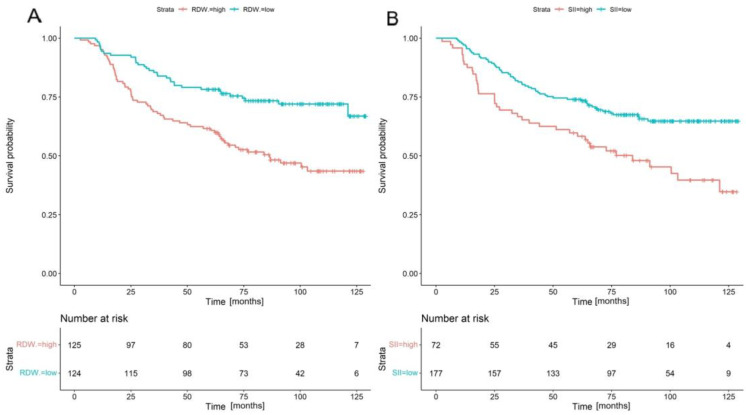
Overall survival curves for RDW (**A**) and SII (**B**) with risk stratification of 249 patients treated with definitive chemoradiation therapy for cervical cancer.

**Table 1 cancers-16-01542-t001:** Clinical characteristics of 249 patients treated with definitive chemoradiation therapy for cervical cancer.

	RDW		SII	
Parameter	Low ≤ 13.4% *n* = 124 (%)	High > 13.4% *n* = 125 (%)	Low ≤ 986.01 *n* = 177 (%)	High > 986.01 *n* = 72 (%)
Age [years]	57.8 (51.5–64)	56.9 (48.5–65.4)	58.6 (52.1–65.5)	51.8 (44.1–58.9)
Histopathology				
SCC	117 (94.4%)	120 (96%)	169 (95.5%)	68 (94.4%)
Adenocarcinoma	7 (5.6%)	5 (4%)	8 (4.5%)	4 (5.6%)
FIGO stage *				
IB1				
IB2		1 (0.8%)	1 (0.6%)	
IB3				
IIA1				
IIA2		1 (0.8%)	1 (0.6%)	
IIB	18 (14.5%)	15 (12%)	31 (17.5%)	2 (2.8%)
IIIA				
IIIB	51 (41.1%)	44 (35.2%)	75 (42.4%)	20 (27.8%)
IIIC1	55 (44.4%)	58 (46.4%)	66 (37.3%)	47 (65.3%)
IIIC2		3 (2.4%)	3 (1.7%)	
IVA		3 (2.4%)		3 (4.2%)
TNM stage *				
IB1	6 (4.8%)	2 (1.6%)	7 (4%)	1 (1.4%)
IB2	2 (1.6%)		1 (0.6%)	1 (1.4%)
IIA1	11 (8.9%)	10 (8%)	19 (10.7%)	2 (2.8%)
IIA2	21 (16.9%)	20 (16%)	25 (14.1%)	16 (22.2%)
IIB	63 (50.8%)	70 (56%)	99 (55.9%)	34 (47.2%)
IIIA	4 (3.2%)		3 (1.7%)	1 (1.4%)
IIIB	8 (6.5%)	12 (9.6%)	12 (6.8%)	8 (11.1%)
IV	9 (7.3%)	11 (8.8%)	11 (6.2%)	9 (12.5%)
ECOG				
0	94 (75.8%)	93 (74.4%)	135 (76.3%)	52 (72.2%)
1	28 (22.6%)	30 (24%)	40 (22.6%)	18 (25%)
2	2 (1.6%)	2 (1.6%)	2 (1.1%)	2 (2.8%)
Positive nodal status	55 (44.4%)	63 (50.4%)	69 (39%)	49 (68.1%)
Radiation modality				
EBRT	1 (0.8%)	2 (1.6%)	2 (1.1%)	1 (1.4%)
EBRT + BT	123 (99.2%)	123 (98.4%)	175 (98.9%)	71 (98.6%)
Concurrent chemotherapy	115 (92.7%)	110 (88%)	159 (89.8%)	66 (91.7%)
Number of cycles				
0	9 (7.3%)	15 (12%)	18 (10.2%)	6 (8.3%)
1	2 (1.6%)	3 (2.4%)	4 (2.3%)	1 (1.4%)
2				
3	6 (4.8%)	3 (2.4%)	7 (4%)	2 (2.8%)
4	10 (8.1%)	9 (7.2%)	15 (8.5%)	4 (5.6%)
5	26 (21%)	31 (24.8%)	41 (23.2%)	16 (22.2%)
6	71 (57.3%)	64 (51.2%)	92 (52%)	43 (59.7%)
RDW [%]	12.9 (12.6–13.2)	14.4 (13.9–15.7)	13.3 (12.8–14.1)	14.1 (13.2–15.9)
NLR	2.38 (1.7–3.35)	2.65 (1.71–3.73)	2.05 (1.53–2.63)	3.96 (3.3–4.84)
PLR	131.76 (106.14–177.6)	160.07 (112.9–218.12)	122.62 (96.65–154.12)	216.78 (172.66–261.29)
SII	602.82 (392.04–868.02)	766.83 (391.64–1187.58)	497.32 (337.39–741.41)	1394.39 (1120.5–1777.18)
RBC [10^6^/μL]	4.45 (4.2–4.7)	4.44 (4.18–4.63)	4.49 (4.26–4.74)	4.22 (3.99–4.53)
HGB [g/dL]	13.7 (12.95–14.2)	13 (11.2–13.9)	13.7 (12.98–14.3)	11.9 (10.6–13.2)
WBC [10^3^/μL]	7.44 (5.89–9.16)	7.36 (6.12–9.27)	6.89 (5.66–8.21)	9.56 (7.73–11.87)
LYMPH [10^3^/μL]	1.91 (1.5–2.36)	1.85 (1.56–2.27)	1.96 (1.63–2.35)	1.72 (1.43–2.2)
NEU [10^3^/μL]	4.63 (3.54–5.91)	4.67 (3.35–6.12)	3.89 (3.17–4.99)	6.6 (5.46–8.51)
PLT [10^3^/μL]	263 (219–305)	283 (230–358)	248 (208.8–284.3)	361 (308–437)

Continuous variables are presented as median and interquartile range (IQR) unless indicated otherwise. * TNM stage was re-evaluated based on diagnostic imaging (MRI, CT, PET-CT); FIGO stage re-evaluation included an additional physical examination. Abbreviations: SCC—squamous cell carcinoma, ECOG—Eastern Cooperative Oncology Group, EBRT—external beam radiotherapy, BT—brachytherapy, RDW—red cell distribution width, NLR—neutrophil to lymphocyte ratio, PLR—platelet to lymphocyte ratio, SII—systemic immune-inflammation index, RBC—absolute red blood cell count, HGB—haemoglobin concentration, WBC—absolute white blood cell count, LYMPH—absolute lymphocyte count, NEU—absolute neutrophil count, PLT—absolute platelet count.

**Table 2 cancers-16-01542-t002:** Cox proportional hazards regression analysis for overall survival in 249 patients receiving definitive chemoradiation therapy due to cervical cancer.

Variable	Univariate Analysis		Multivariable Analysis	
	Hazard Ratio (95% CI)	*p*-Value	Hazard Ratio (95% CI)	*p*-Value
Age [years]	0.99 (0.973–1.008)	0.270	0.997 (0.978–1.016)	0.607
Histopathology (SCC vs. adenocarcinoma)	0.953 (0.388–2.344)	0.917	1.14 (0.46–2.84)	0.897
Positive nodal status	1.514 (1.018–2.251)	0.041	1.26 (0.82–1.92)	0.361
ECOG status (1 vs. 0)	0.95 (0.58–1.57)	0.852	0.91 (0.55–1.50)	0.781
ECOG status (2 vs. 0)	4.25 (1.54–11.71)	0.005	4.55 (1.61–12.85)	0.023
FIGO stage (III and IV vs. I and II)	2.184 (1.059–4.505)	0.034	1.68 (0.78–3.63)	0.177
TNM stage (III and IV vs. I and II)	1.589 (0.969–2.541)	0.067		
RBC [10^6^/μL]	0.666 (0.43–1.031)	0.069		
RDW	1.183 (1.082–1.294)	<0.001	2.04 (1.32–3.16)	0.001
HGB [g/dL]	0.866 (0.77–0.975)	0.017	0.89 (0.78–1.02)	0.909
WBC [10^3^/μL]	1.112 (1.05–1.179)	<0.001		
NEU [10^3^/μL]	1.138 (1.063–1.219)	<0.001		
LYMPH [10^3^/μL]	1.08 (0.802–1.454)	0.611		
PLT [10^3^/μL]	1.002 (1–1.004)	0.107		
NLR	1.243 (1.082–1.428)	0.002		
PLR	1.002 (0.999–1.004)	0.199		
SII	1 (1–1.001)	0.001	1.42 (0.89–2.25)	0.138

In the multivariate analysis, the RDW and SII were considered as dichotomous variables with cut-off values of 13.4% and 986.01, respectively. For the variables remaining in the multivariate model, hazard ratios with a 95% CI were reported. Abbreviations: CI—confidence interval, HR—hazard ratio, SCC—squamous cell carcinoma, ECOG—Eastern Cooperative Oncology Group, RDW—red cell distribution width, NLR—neutrophil to lymphocyte ratio, PLR—platelet to lymphocyte ratio, SII—systemic immune-inflammation index, RBC—absolute red blood cell count, HGB—haemoglobin concentration, WBC—absolute white blood cell count, LYMPH—absolute lymphocyte count, NEU—absolute neutrophil count, PLT—absolute platelet count.

## Data Availability

Anonymized data are only available upon request due to privacy and ethical restrictions.

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
