# Peer review of "The Prognostic Value of the Systemic Immune-Inflammation Index (SII) and Red Cell Distribution Width (RDW) in Patients with Cervical Cancer Treated Using Radiotherapy"

_cancers, 2024, doi:10.3390/cancers16081542_

Round 1

Reviewer 1 Report

Comments and Suggestions for Authors

In the article "The Prognostic Value of the Systemic Immune-Inflammation Index (SII) and Red Cell Distribution Width (RDW) in Radiotherapy-Treated Cervical Cancer Patients" Staniewska et al. analyzes the prognostic biomarker value of SII and RDW in cases of cervical cancer treated with radiotherapy. The authors identify optimal cut-off values for SII and RDW and present the impact of including RDW in multivariable models by decreasing the Akaike information criterion. However, after a series of details about the treatment, no explicit comment appears later. We anticipate that I will identify an estimate of EQD2 dose (EBRT+BT) and an inclusion of the cumulative dose of Cisplatin in the analysis. I do not necessarily ask the authors to perform such an analysis, although I consider it interesting, but if these data are not subsequently analyzed, the information about radiotherapy and brachytherapy is far too exhaustive. If we analyze the brachytherapy technique and dose prescription, a simple comparison between prescription brachytherapy at point A and imaging-guided brachytherapy would be relevant if it exists. If we detail so much about brachytherapy, we must also detail EBRT. Was lumboaortic irradiation performed? Was bone marrow sparing used? you probably don't have the data.. in this case obviously the data about the brachytherapy prescription are not relevant because we cannot get a complete picture and we cannot propose a division of cases by treatment. It is relevant for a possible dynamic of NLR/SII. In the conclusions, SII is not mentioned at all. I congratulate the authors for the idea, but the article needs to be expanded or reduced by irrelevant information for the analysis.

Author Response

Dear Reviewer,

Thank you for your thorough review of our work, and valuable comments. We have made several adjustments described below, which improved the quality of the manuscript.

Following your advice, we decided to shorten the procedural description of radiotherapy, and moved most of the details to a Supplementary File. We have also provided median and interquartile range of EQD2 (EBRT+BT). However, considering that all patients received doses in the curative range, we prefer to not add these values to the analysis. We agree that the complexity of the treatment adds significant layers of confounding, and estimation through just one value (EQD2) would be too much of an approximation.

Considering the procedural details, neither lumboaortic irradiation, nor bone marrow sparing was performed. All patients received elective pelvic lymph node irradiation either up to the level of L4/L5 vertebral junction, or up to the aortic bifurcation, depending on attending physician’s decision. We have added a short comment to the manuscript.

With few exceptions of minor dose reductions, all patients in our study received weekly Cisplatin dose of 40 mg/m2. We have added the number of weekly cisplatin cycles received by the patients to the manuscript.

Finally, we agree that it would be better to comment on all evaluated indices, and we have adjusted the conclusions.

Reviewer 2 Report

Comments and Suggestions for Authors

The major purpose of the manuscript presented by Staniewska et al. was to evaluate a predictive value of red cell distribution width (RDW) or systemic immune inflammation index (neutrophils-to-lymphocyte ratio (NLR) and platelets-to-lymphocyte ratio (PLR) on clinical endpoint overall survival in 249 patients with cervical cancer treated with definitive chemoradiation therapy. In more details, authors indicate a low RDW (≤13.4%) and low SII (≤986.01) to be associated with a longer overall survival with RDW to remain an independent prognostic factor in multivariable analyses.

Although the study covers an interesting topic of easy-to-handle prognostic/predictive markers in cervical cancers, there are several conceptual problems, lack of novelty and a weak performance that prohibit the reviewer´s recommendation for publication in Cancers.

Major points of criticism:

1.     Analyses were highly restricted to the clinical endpoints overall survival and lack assessment of more specific oncological endpoints like disease-free survival, and local or distant tumor control/recurrences.

2.     Moreover, authors failed to clearly state novel findings in their analyses.

3.     Authors provided data on external beam radiotherapy (EBRT) brachytherapy schedules in detail, but completely failed to describe techniques in the material and methods section. In line with that, tables are very complex but do not provide any information regarding the analyses with very low patients in some groups and therefore should be omitted.

4.     Authors provided a simple enumeration of patient numbers with a low and high values of RDW and SII but did not indicate a correlation with these histopathological parameters displaying Spearman’s correlation coefficients.

5.     Discussion section is weak in performance and mainly covers an enumeration of findings of comparable studies without discussing their findings. How do authors mechanistically explain an association of RDW with overall survival?

6.     Moreover, authors failed to clearly state novel findings in their analyses and did not quote and discuss a multitude of manuscripts on the topic including Lu Z et al. Front Oncol 2020;10:341, Guo H et al. J Inflamm Res 2023;16:5145-5156; Mleko M. et al. Cancer Manag Res 2021;13:5491-5508; and Ferioli M. J Pers Med 2023;13(8):1229.

Author Response

Dear Reviewer,

We thank you for your comments and critical review of our work. We understand the Reviewer’s concerns; however, we are convinced that our work provides a valuable addition to the field, and is worth considering. Below, we have provided responses to the major points of criticism provided by the Reviewer:

Ad 1.

In the setting of retrospective analysis, we believe that the overall survival (OS) is the least biased endpoint. Importantly, the use of census data allows for longer follow-up and no (informative) censoring. Time-to-event endpoints such as DFS also suffer from the impact of uneven follow-up intervals on censoring when collected retrospectively. Finally, OS is ultimately the most important measure for most patients undergoing oncologic treatment for advanced cancers. Considering the methodologic limitations, we are not convinced that reporting of additional intermediate clinical endpoints would add value to our study.

Ad 2.

We found that RDW is an cheap and widely available index independently associated with survival in patients treated with chemoradiotherapy for cervical cancer, and could be used to improve the accuracy of pre-treatment prognostic assessment. This could allow for treatment intensification in patients at high risk of failure; however, prospective trials are necessary to verify this hypothesis. Conversely, no clear association could be identified for SII. We apologize to the Reviewer for indecisive stating of findings; we have improved the conclusions of our manuscript.

Ad 3.

We thank the Reviewer for this comment. Indeed, on second though we agree to move the details into the Supplement, reduce the manuscript volume, and clarify the methods description.

Ad 4.

Following the reviewer’s comments, we performed analyses for correlations between evaluated indices and clinical characteristics of the patients, and added them to the results section.

Ad 5.

We thank the Reviewer for this thoughtful comment. There are several mechanisms in which RDW can affect survival; We mentioned this topic in the Introduction section. RDW represents the red blood cell (RBC) volume variation. This marker reflects the inflammatory response related to the accelerated cancerogenesis and neo-angiogenesis induced by cytokines produced by cancer cells. Therefore, elevated values are associated with inflammation related to cancer and the blood loss associated with more advanced disease. Higher clinical stage of the cancer connected with more intensive inflammation, reflected by RDW value, is associated with worse survival of patients.

Ad 6.

Thank you, we have added relevant mentioned references to the manuscript in the introduction and discussion sections. 

Reviewer 3 Report

Comments and Suggestions for Authors

The authors performed a retrospective analysis to evaluate the association between overall survival (OS) in 249 patients treated by chemoradiotherapy for cervical cancer and pretreatment levels of red blood cell distribution width (RDW) and systemic immunoinflammation index (SII). The blood samples were collected before treatment for histopathologically confirmed cervical cancer at stages IB2 to IVA, according to the FIGO 2018 system. The overall aim was to determine if RDW and SII could be used as independent prognostic factors to assess patient survival.

The results were generally well presented and support the conclusion that RDW and SII have a predictive value. Their results support those published by other teams. An important question will be to determine which cut-off of RDW and SII to use n clinic.

 Comments

1)      Lines 31-32: The authors are invited to clarify this sentence: “The RDW remained as a remained as an independent prognostic factor in the multivariable model”.

2)      The radiotherapy plans were well presented and quite similar from patient to patient.

3)      It would be appropriate to add a table reporting the chemotherapy treatment of these patients.

4)      Figure 1A and B: The authors are invited to clarify the meaning of “Density”.

5)      Figure 2: The authors are invited to add which unit was used for “Time”.

6)      Correlation between RDW and SII levels with OS (Figure 2) could have been performed with subgroups of patients based on the disease stage for those whose patient numbers allow for valid statistical analysis.

7)      The authors are invited to better define “Akaike Information Criterion” and its clinical usefulness. What a low or high Akaike Information Criterion mean? Is a decrease from 1028.25 to 1018.15 significant?

8)      Why the predictive value of RDW was not mentioned in the conclusion.

Author Response

Dear Reviewer,

We would like to thank you for your thorough review, and valuable comments.

Ad 1.

We thank the reviewer for pointing out this error, we have removed the unnecessary repetition in the manuscript.

Ad 3.

We thank the reviewer for this comment. All patients received concomitant Cisplatin in a dose of 40mg/m2. We have added this information to the manuscript, along with information regarding received number of concurrent chemotherapy cycles.

Ad 4.

“Density” means representation of distribution of RDW/SII; a mixture model of Gaussian distributions is fitted to the histogram of the biomarker. The optimal cut-off is determined as the value where the probability density functions of the mixing distribution coincide. We have added a short explanation to the Figure 1A/B.

Ad 5.

Thank you for noticing this error, we have added the missing time unit [months] to the Figure 2.

Ad 6.

The patient’ stage (FIGO and TNM) was included into the analysis. However, we do not agree that it is possible to perform the correlation between RDW and SII levels with OS using our dataset, with subgroups based on the diseases stage, due to low number of patients in the subgroups. Considering that the analyses would not have sufficient power, we would prefer to omit them.

Ad 7.

Akaike information criterion is an estimator of prediction error and as such, lower values indicate a better fit of the model. A decrease from 1028.25 to 1018.15 corresponds to a relative likelihood of a model of 0.0064; in other words, the worse model (with AIC 1028.25) is 0.0064 times as likely as the model with AIC 1018.15 to minimize information loss. Therefore, at alpha=0.05, this difference corresponds to a statistically significant improvement in model predictions. Including blood test indices into the model allow to predict the patient’s prognosis with higher accuracy.

Ad 8.

We have added the information about the prognostic value of RDW in the conclusion, similarly to the SII.

Reviewer 4 Report

Comments and Suggestions for Authors

The search and substantiation of prognostic factors in oncology is one of the burning topics. This retrospective study is one of the necessary links in the search for such markers. The authors use adequate statistical methods and an acceptable number of parameters in attempt to determine the predictive level of SII and RDW indicators. However, it is obvious that this research is not final. It is advisable to continue/expand the study, supplementing it with such indicators as concomitant diseases, results of histological studies, etc. I hope this will be the authors' next paper. 

Comments on the Quality of English Language

It is advisable for authors to carefully proofread the text, correcting some technical and stylistic inaccuracies. 

Author Response

Dear Reviewer,

We thank you for your thorough review, and valuable comments.

Round 2

Reviewer 1 Report

Comments and Suggestions for Authors

From my point of view, my objections and recommendations were respected in the revised version of the manuscript

Reviewer 2 Report

Comments and Suggestions for Authors

I see my prevoius comments adequately addressed.